# Performative Structural Design Optimization: Generative Algorithm for a Preliminary Study of a Voided Beam

Laura Sardone [1], Alessandra Fiore [1,*], Amedeo Manuello [2] and Rita Greco [3]

1 Politecnico di Bari, DICAR, Via Orabona 4, 70125 Bari, Italy
2 Politecnico di Torino, DISEG, Corso Duca degli Abruzzi 24, 10129 Torino, Italy
3 Politecnico di Bari, DICATECh, Via Orabona 4, 70125 Bari, Italy
* Correspondence: alessandra.fiore@poliba.it

**Abstract:** In the world of structural design, in most cases, there is a need to control the shape of structural elements and—at the same time—the performance that each one can achieve. With the evolution of structural analysis tools, nowadays it is possible not only to have an immediate investigation of the structure's performance, but also to search for the best shape by imposing geometric constraints. The aim of this paper is to present an innovative methodology called the performative structural design optimization (PSDO) method, based on the use of algorithm-aided design (AAD). The proposed approach deals with an emptied voided beam; starting from the parameterization of a large-span beam, the search method for the most performing shape is accomplished by multi-objective evolutionary algorithms (MOEAs). The obtained results are characterized by a double optimization: the structure achieved by the hypervolume estimation algorithm for multi-objective optimization (HypE Reduction) (OCTOPUS) represents the starting shape for the application of form-finding, giving so the possibility to obtain different feasible solutions from a single study and to choose the best one in terms of structural behavior.

**Keywords:** parametric design; PSDO method; visual programming; computational geometry; structural optimization; form-finding; hypervolume indicator; multi-objective optimization; conceptual design



## 1. Introduction

Searching for the "best solution" is a crucial point of many, maybe all, human activities that can be roughly summarized as "do more with less". Within all the possible formal descriptions dealing with the decision-making, a quite useful one is to delineate it as finding the best solution in certain circumstances to achieve a particular goal from multiple alternatives [1].

Different disciplines are intrinsically questioned and involved in the design phases in the architecture field [2] so furnishing useful parameters for a correct design.

Together with other parameters, each discipline contributes to designing spaces, shapes, and structures; one of the main questions is the search for alternatives, that is, the exploration of alternatives for solving problems discovered at each design stage. However, as there is a limit in the ability to gather information, only some alternatives are feasible: (i) alternative evaluations, which means predicting the outcome of each search alternative, to which comparison and evaluation follow. However, due to the limitation in evaluation capacity, it is impossible to fully predict and evaluate all results; (ii) alternative selection: after anticipating the results of each alternative, by comparison, and evaluation, it consists in selecting the relatively best alternative that satisfies a certain objective.

For these reasons, a critical step for an optimal design is to execute a reliable "initial" solution that summarizes all the design criteria to be executed up to the final phase.

One of the most critical phases in construction design—commonly called "Conceptual Design"—is the first step of architectural design, in which the bases for obtaining

the final product are pronounced. In this preliminary phase, the decision of the performative response of the designed structure is intrinsically instilled which, in most cases, strictly depends on the shape—combined with the material properties—that we attribute to structural elements.

In this field, visual programming (VP) is attracting particular interest for many architecture and civil engineering applications. VP languages represent an alternative to traditional text-based programming approaches and consist of graphical methods based on the use of blocks, also called nodes [3]. The block input fields contain the problem parameters, while the output fields provide their results. Grasshopper for Rhinoceros 3D, Dynamo for Autodesk Revit, Allplan Visual Scripting for Nemetschek Allplan, and Marionette for Vectorworks are the most common VP tools [4].

VP scripts are mainly adopted to automatize parametric geometry modeling [5] but can also be implemented for other engineering tasks, such as for energy and thermal analyses in buildings [6,7].

Merged with an optimization algorithm [8,9], they create an automated design tool and generative design, in which a user selects the constraints and ranges of involved parameters, while the algorithm optimally adjusts the values. Within this framework, some researchers have recently advanced the use of VP methodologies for solving volume, weight, and cost minimization problems. For instance, Aydın and Ayvaz [10] adopted a VP approach to minimize the total cost of prestressed concrete beams by adjusting their shape, prestressing, and arrangement. Another application of VP was proposed by Sardone et al. [11], in such a case to minimize the volume of variable cross-section beams through the use of the Gh-Octopus solver, in the Grasshopper environment. The VP technique was also implemented by Lee [12] for finding the optimal shape of arch structures, by using the plug-in Karamba3D in the Grasshopper environment and the plug-in NM-opti.

In this scientific contribution, a VP algorithm has been implemented to create a new preliminary design methodology, by exploiting the latest architectural and structural design technologies [13].

The presented approach adopts the following tools: (i) Rhinoceros 3D—software responsible for visualizing the structural element; (ii) Grasshopper—utilized for the parameterization of geometries; (iii) Karamba3D—plug-in developed to retrieve the results of the Finite Element Analysis (FEA); (iv) Octopus—Hypervolume Estimation Algorithm for Multi-Objective Optimization [14]; (v) Kangaroo—useful for the application of form-finding.

The aim of the method is to speed up the calculation process through the search for form—which today is also closely linked to aesthetic study in the architectural field—and structural performance. To achieve a performative structure, computational optimization techniques have become a necessity during the conceptual design phase. In the following subsections, the benefits of pre-optimization are explored by introducing the performative structural design optimization (PSDO) approach. The case study describes a central structural design case, that is a horizontal beam defined in a rectangular domain that is "emptied" and "voided" to reduce its weight/volume without excessive loss of stiffness and strength. The chosen case study represents one of the most common problems in structural engineering, also treated in a continuous and discrete configuration.

### 1.1. Environmental and Economic Advantages: The Need for Structural Optimization

Due to its availability, durability, and safety, concrete has become the dominant building material around the world. However, high demand needs to be mitigated to save costs and preserve the environment from natural resource depletion and $CO_2$ emissions in the concrete manufacturing process [15]. On a world scale, according to the latest available data, cement production is close to two billion tons per year, and, by 2050, demand will grow, reaching four times more than in 1990. Furthermore, cement production is closely linked to demand for steel; in 2016, Asia and Oceania alone required 1000 million tons of steel [16]. These data, translated in terms of environmental impact, mean producing disastrous consequences on the environment.

The reduction in the production of building material aims at preserving the environment and at reducing the cost of construction itself. However, each structural element, to be optimized, should not show structural adequacy defects. For example, spandrel beams and bridges' exterior girders are designed to withstand the combination of bending, shear, and torsion stresses that are more dangerous than any other stress. When a solid reinforced concrete beam is subjected to combined moments and bending moments prevail over torsional moments (M/T ≥ 1.7), the beam can be stronger than its pure bending strength or pure torsion strength. So, increasing the torsional moment can improve the stiffness [17].

Compared to a solid beam, a voided reinforced concrete beam could appear structurally less performing, especially with regard to bending moment. A solid structure could seem preferable to ensure adequate resistance to the fragility phenomena due to its concrete nature. In this framework, this contribution aims at demonstrating how a performative double-step optimization process (PSDO) could reduce the amount of cement and provide adequate strength, especially in the context of flexural behavior investigation.

### 1.2. Voided Arch-Shaped Box Girder

The conceptual inspiration of this case study is the search for shape from Robert Maillart, in the field of bridge design. Maillart brought a revolution in the world of construction by introducing the idea of voided beams for main girders, obtained thanks to his structural intuition without any automatic analysis support. In more detail, from 1900, Maillart designed his first bridge in the village of Zuoz, a 30 m single-span lowered arch bridge. The bridge became the precursor of modernity, being the first reinforced concrete box girder bridge in history (Figure 1a). The idea was born from the desire to combine the aesthetic elegance of the arch with a shapeless and ductile material as reinforced concrete. The load-bearing capacity of the bridge did not rely entirely on the arch but required abutments and slabs, and for this reason, a box girder was created and used for the construction. However, the bridge became a source of discussion in 1903, when dangerous cracks appeared in the sidewalls connected with the abutments. Although the phenomenon was called not dangerous by Maillart itself (mainly due to shrinkage and thermal effects), this event made him lose the patent for the Box Girder. To solve the problems due to creep effects and to reduce the cost of construction, Maillart took up the elegance and geometry of Zuoz Bridge, increasing the span size (from 30.0 m to 51.25 m) and the height of the arch at mid-span (about 5.7 m above the ground). The improvement started from those parts where he had more problems in the previous project: Maillart began a study to remove structural parts and material, creating the Tanavasa bridge, an extraordinarily light and performative structure with fewer costs. The bridge was built in one year using formwork (mold) for concrete, and it cost only CHF 28,000 [18]. Furthermore, the shape of the voids (Figure 1b) represented a solution to ensure a modern aesthetic as well as the discontinuity provided a sufficient structural strength.

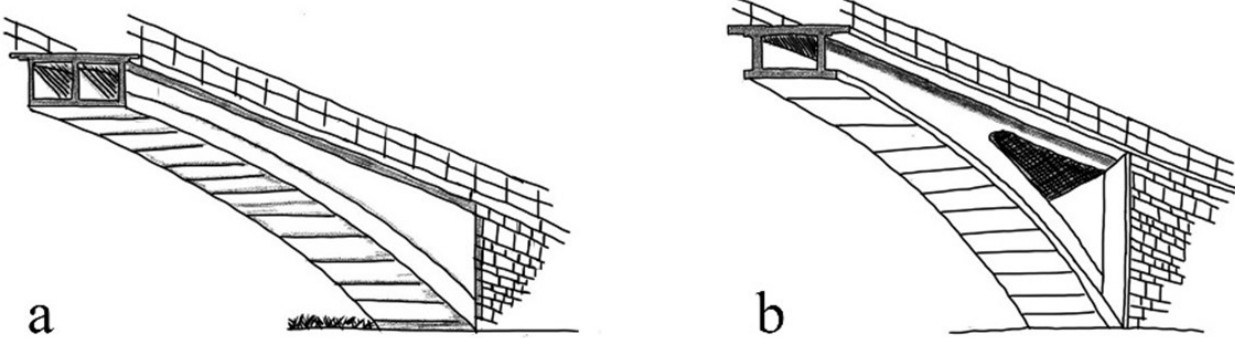

**Figure 1.** (**a**) Zuoz Bridge cross-section; (**b**) Tavanasa Bridge cross-section.

The bridge collapsed in September 1927 due to a landslide and subsequent studies carried out by Prof. Mirko Ros (Zurich, 1937) proved that the mechanical properties of

involved materials were not deteriorated, so confirming that the collapse was exclusively caused by the landslide effects [19].

In this framework, the aim of the proposed study is to analyze voided arch-shaped concrete beams, focusing on the effects of geometry on structural efficiency (and reducing material waste). Their structural behavior is investigated by the PSDO method.

### 1.3. Merging MOOPs and Form Finding for a Performative Concrete Shell Structure

The elegant complexity of structural and architectural shapes obtained through the form-finding method has become a broad prerogative for designers who try to merge structural stability with the compositive beauty of the architectural design. Thanks to additive manufacturing connected to 3D printing—which has become part of the construction industry—nowadays, free-form shape structures are allowed in the actual construction. Free-form shells, through proper design, allow us to obtain high-performance structures with maximum material savings, leading to projects characterized by wide spans thanks to their weight-to-height ratio and high rigidity. Furthermore, shell structures are spatially curved structures that can be defined as "form resistant structures" [20]. By introducing steel and concrete into 3D printing, the opportunity to define structures with complex geometries—with the possibility of exploring new shapes by exploiting free form shells— is increased.

A structure obtained through form-finding can be differently designed according to the needs given by the specific project: a membrane can react purely to compression or purely to tensile stresses, depending on the forces applied in the design phase. The very low bending gives the advantages derived from the use of membrane within the structure. However, concrete shells, in particular, bending can cause tension that accumulates in extreme fibers, leading to cracking. Furthermore, the material nonlinearity, in combination with the wrong geometry adopted for the structure, can cause failures giving the shell the characteristics of an "imperfection-sensitive structure" [21].

Shells represent a great opportunity to produce innovative and sustainable architectures allowing the minimum weight of building elements. However, since shells are—by definition—sensitive to imperfections, it is clear how certain criteria regarding architectural shape must be respected. Therefore, a numerical model assisted by a two-phase optimization algorithm has been implemented: the result of shape optimization is directly connected with the solver adopted for the form-finding in order to have doubly performing results, reducing the calculation times, and providing high accuracy in the optimization process thanks to the FEA.

This work aims at proposing a method to design—debating the challenges/advantages in designing—a thin concrete shell structure (50.0 m × 10.0 m) and at investigating the efficiency of a thin continuous shell proposed as a large-span beam. The adopted methodology is based on: (i) 3D modeling in Rhinoceros 3D; (ii) finite element modeling in Gh-Karamba 3D; (iii) shape optimization in Gh-Octopus, based on hypervolume estimation algorithm for multi-objective optimization problems (MOOPs); (iv) a double-step optimization connecting Octopus results with Gh-Kangaroo, by which the last optimization, based on form-finding method, will be performed.

## 2. Introduction to Performative Structural Design Optimization Method

In this paper, the PSDO method is approached to find the best shape of structural elements and simultaneously ensure structural safety and best performances. PSDO consists of an iterative method based on performative computational architecture (PCA) [22,23], structural performance evaluation, and optimization.

Specifically, the presented methodology is composed of the following steps (Figure 2):

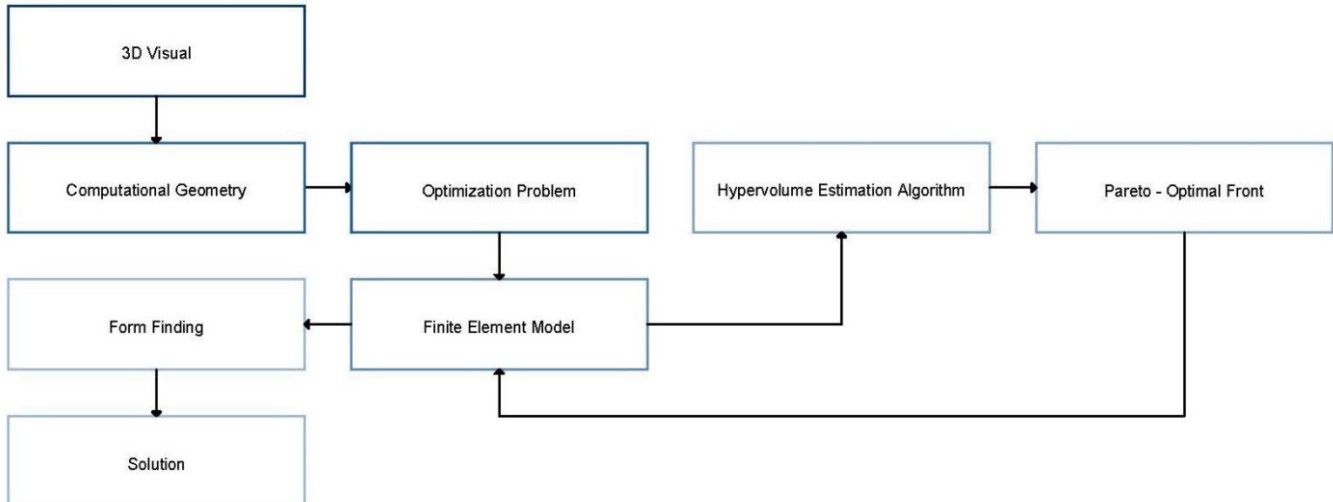

**Figure 2.** Tools workflow.

Step 1: visualization of the structural element by the software Rhinoceros 3D;

Step 2: parameterization of geometries through the VP software Grasshopper;

Step 3: retrieval of the results of the FEA by the plug-in Karamba3D;

Step 4: Multi-Objective Optimization by the hypervolume estimation algorithm Octopus [14].

Step 5: import and evaluation of the optimized geometry with the application of form-finding by the plug-in Kangaroo.

More precisely, the computational geometry methodology allows for the generation of geometric models through the parameterization of curves, successively analyzed by a FEA. A third step, which emerged from the FEA results, is represented by the assessment of structural performances. The best performance must be able to respond to objectives (or pre-established constraints) and ensure feasibility, by including the load and resistance factor design (LRFD) and the serviceability limit state (SLS). The optimization phase considers metaheuristic algorithms to discover optimal design solutions within a systemic research process, checking the static and dynamic feasibility. The final phase defines the nearly optimal architectural and engineering solution.

### 2.1. Computational Design Using Grasshopper

The proposed methodology starts from a solid beam geometry (Figure 3) whose dimensions are defined as follows: $L$ (span-length) = 50 m; $h_e$ (depth) = 10 m; $b$ (shell cross-section) = 0.35 m.

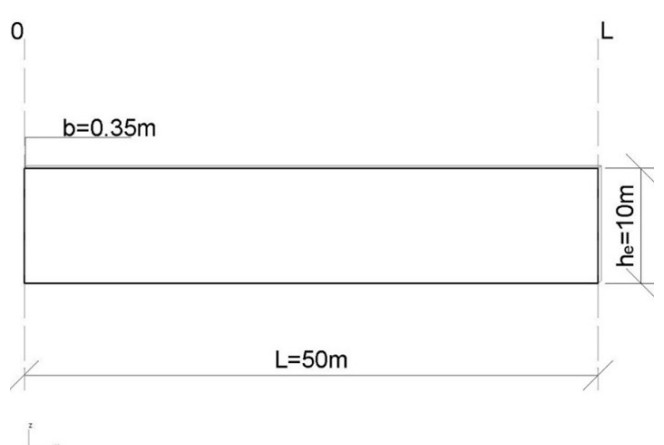
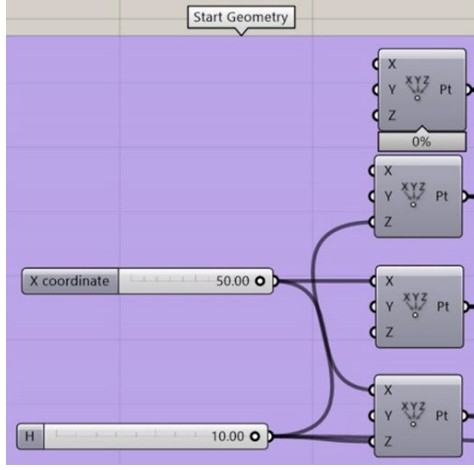

**Figure 3.** Test case—start geometry.

The just-described parameters represent the constant dimensions of the starting geometry implemented in Grasshopper [24] (Figure 3).

The considered shape is described by means of a set of physical parameters:

$$\{p_i\}_{i=1,\cdots,N} \tag{1}$$

including the optimization variables. In this case, the solver can act by emptying the solid beam, in the respect of certain parameters that, for the test case, also represent geometrical constraints and reflect the preliminary creative idea on the basis of which the conceptual design develops.

The first emptying function is retrieved starting from the general circumference equation, centered with respect to the *z*-axis:

$$x^2 + y^2 + \beta z + c = 0 \tag{2}$$

It is worth noting that, for the bottom profile of the beam, a circumference arc shape, that is a constant curvature outline, is preferred, at the aim to simplify the construction stage, also with reference to a possible prefabrication system, and to reduce costs.

The arc described by Equation (2) can be parameterized by imposing its passage through three points (Figure 4), named $P_1, P_2, P_3$, characterized by the following coordinates:

$$P_1 = (-L, 0)$$

$$P_2 = (0, h_{m'})$$

$$P_3 = (L, 0)$$

where $h_{m'}$ represents the value of the emptying function:

$$h_{m'} \in [0, h_e - \delta] \tag{3}$$

where $h_e$ is the initial beam height and $\delta \cong h_e/10$ a fixed minimum value of the depth.

The beam section depth function is defined as:

$$h_m = h_e - h_{m'} \tag{4}$$

that, by substituting the emptying function $h_{m'}$ with Equation (2) and assuming $\beta > 0$ and $c < 0$, becomes:

$$h_m = h_e - \left( x^2 + y^2 + |\beta| z - |c| \right) \tag{5}$$

So, the emptying function $h_{m'}$ is expressed as:

$$x^2 + y^2 + |\beta| z - |c| \tag{6}$$

and represents a first geometrical constraint.

Starting from the curve in Equation (6), a second curve is defined, passing through the points $P_{1'}, P_{2'}, P_{3'}$ and representing the basis to create the second geometrical constraint in the optimization problem, that is the insertion of voids in the beam (Figure 5):

$$(x^2 + y^2 + |\beta| z - |c|) \in [P_{1'},\ P_{2'},\ P_{3'}] \tag{7}$$

where:

$$P_{1'} = \left( -L, \frac{h_e}{2} \right) \ \Rightarrow \ \frac{h_e}{2} \ (const.) \tag{8}$$

$$P_{2'} = \left( 0, \frac{h_m}{2} \right) \ \Rightarrow \ \frac{h_m}{2} = \frac{h_e - h_{m'}}{2} \tag{9}$$

$$P_{3'} = \left( L, \ \frac{h_e}{2} \right) \ \Rightarrow \ \frac{h_e}{2} \ (const.) \tag{10}$$

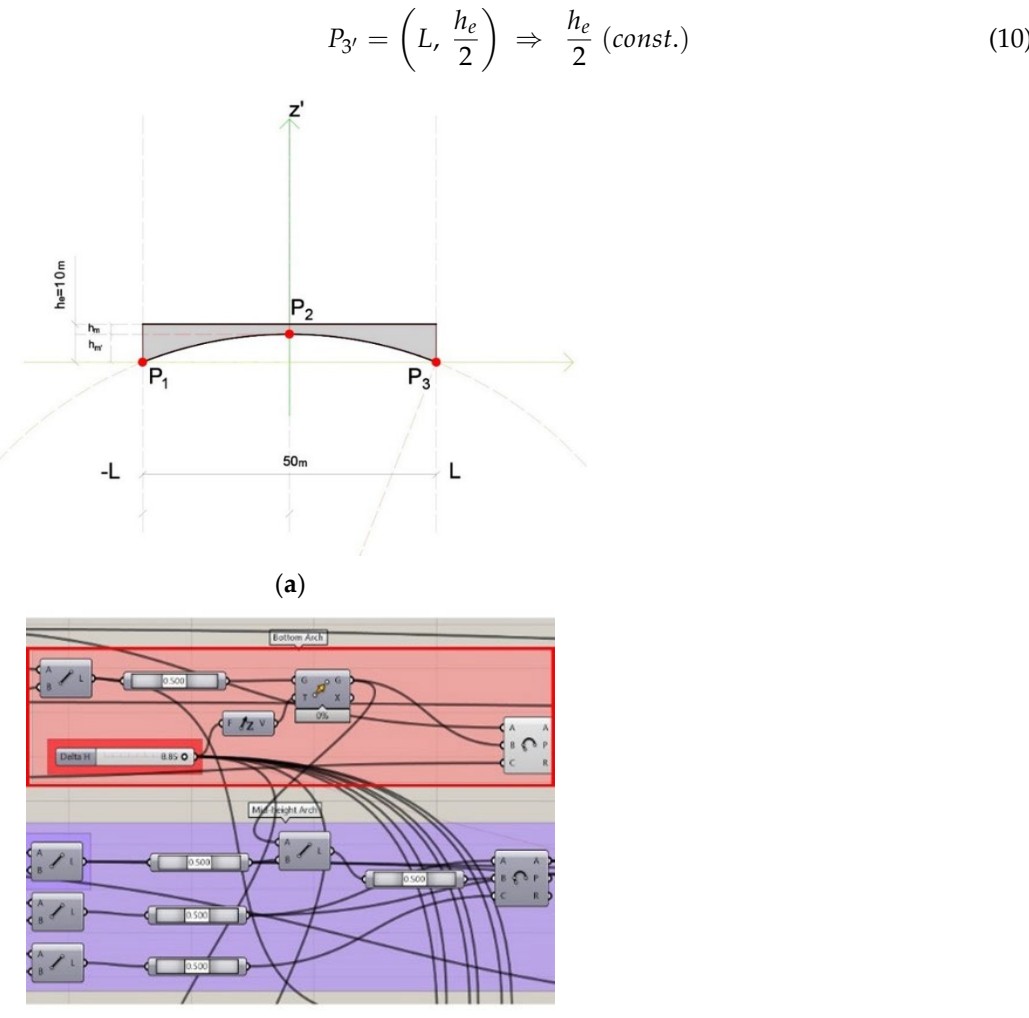

(a)

(b)

**Figure 4.** (**a**) Emptying function—the arc of circumference passing through three points; (**b**) Grasshopper development of the emptying function (upper definition in red).

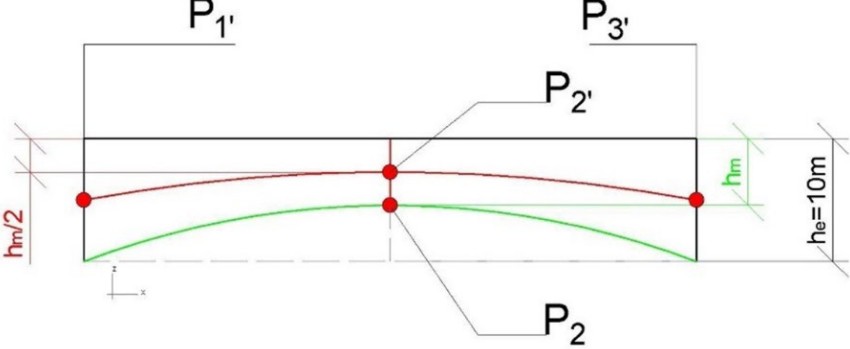

**Figure 5.** Shifted curve (Equation (8)).

In fact, with the aim to achieve both a bigger number of voids to introduce in the beam and more flexibility in their shape, so as to allow a better distribution of stresses, the ellipse shape is selected, under the hypothesis that all the centers of the ellipses are placed on the curve described by Equation (7). Moreover, in order to simplify the problem from a computational point of view, the condition of symmetry is set.

The ellipse equation is so considered:

$$\overline{PF_1} + \overline{PF_2} = 2a \tag{11}$$

that can be rewritten as:

$$\sqrt{(x - x_1)^2 + (y - y_1)^2} + \sqrt{(x - x_2)^2 + (y - y_2)^2} = 2a \tag{12}$$

and that, in parametric form, becomes:

$$\begin{cases} x = a \cos t \\ y = b \sin t \\ 0 \leq t < 2\pi \end{cases} \tag{13}$$

In the case under examination, the $x$-axis is replaced by the curve described in Equation (7) and the $y$-axis is replaced by the $z$-axis.

With reference to the ellipse, it is imposed that: $a$ = horizontal semi-axis; $2a$ = horizontal axis; $b$ = vertical semi-axis; $2b$ = vertical axis.

So, it is possible to recognize the major axis by comparing the terms $a^2$ and $b^2$ as follows:

$$a^2 \geq b^2 \implies \overline{PF_1} + \overline{PF_2} = 2a \tag{14}$$

$$a^2 < b^2 \implies \overline{PF_1} + \overline{PF_2} = 2b \tag{15}$$

Within this configuration, the circumference shape is included by setting:

$$a^2 = b^2 \implies x^2 + y^2 + \alpha x + \beta z + c = 0 \tag{16}$$

In this way, three possible solutions regarding the shape of voids are obtained: ellipses with a major horizontal axis (Equation (14)), ellipses with a major vertical axis (Equation (15)) and circumferences (Equation (16)). The shape of voids belongs to the physical parameters set in Equation (1).

Equation (14) is strictly dependent on the position of point $P_2$ (and on the parameter $h_{m'}$ (Equation (3))). As the height of point $P_2$ increases, the ellipse (or circumference) must maintain a size to ensure a sufficient thickness between the perimeters of the void and of the beam, so providing a suitable rigidity and strength to the structural component.

The first void is fixed on the external edge (Figure 6a,b) with the center placed in $P_{1'}$ (Equation 8). In this case, the domain of parameter $b$ is set as follows:

$$b_1 \in \left[ \frac{0.5 \, h_e}{2} ; \frac{0.8 \, h_e}{2} \right] \tag{17}$$

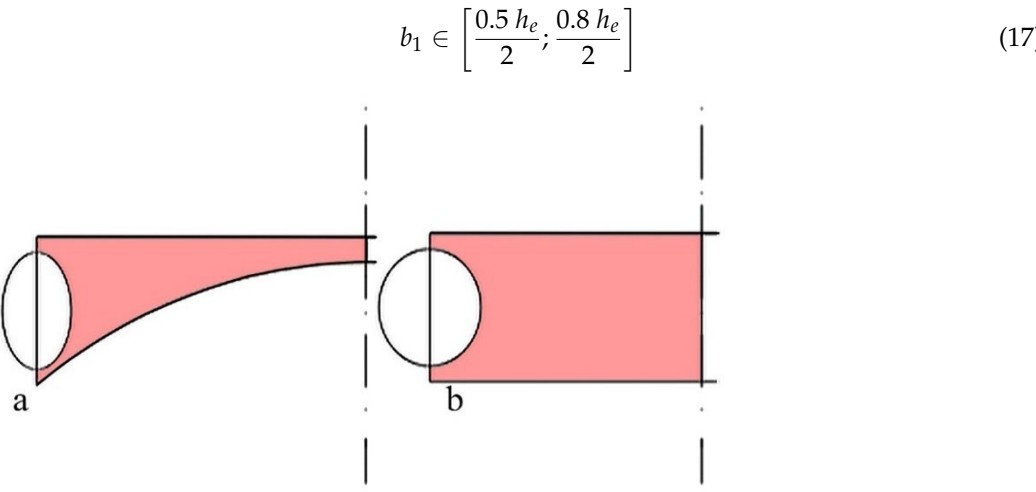

**Figure 6.** (**a**) Edge void no.1 with $h_{m'} > 0$; (**b**) edge void no.1 with $h_{m'} = 0$.

In order to determine the domain of parameter $a$ for the first void, it is necessary to trace the tangency point between the void and the initial curve described in Equation (6) (passing through the points $P_1$, $P_2$, $P_3$). The domain of $a$ is defined by imposing a sufficient distance between the perimeter of the void and that of the beam (Figure 7). By expressing the parameter $a$ as a function of the radius $(d - n)$, $d$ being the distance between $P_{1'}$ and the just introduced tangency point, it is imposed that:

$$a_1 = (d - n); \quad n \in [n_{min}; n_{max}] \tag{18}$$

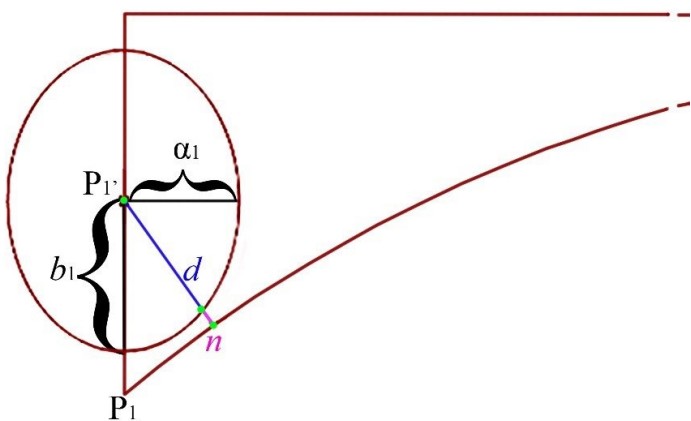

**Figure 7.** Distance between Void N.1 and arch of the circumference.

The following relation between the parameters $a$, $b$, and $h_{m'}$ can so be derived:

$$h_{m'} \in [0, h_e - \delta] \implies a^2 < b^2 \implies \overline{PF_1} + \overline{PF_2} = 2b \tag{19}$$

$$h_{m'} \in [0, 0] \implies a^2 > b^2 \implies \overline{PF_1} + \overline{PF_2} = 2a$$

By using the *PlaneTrimCurve* component (Pufferfish-Grasshopper plug-in), it is possible to trim the curve described in Equation (14), along the *z*-axis, obtaining the first definitive parametric void at the beam extremity (Figure 8).

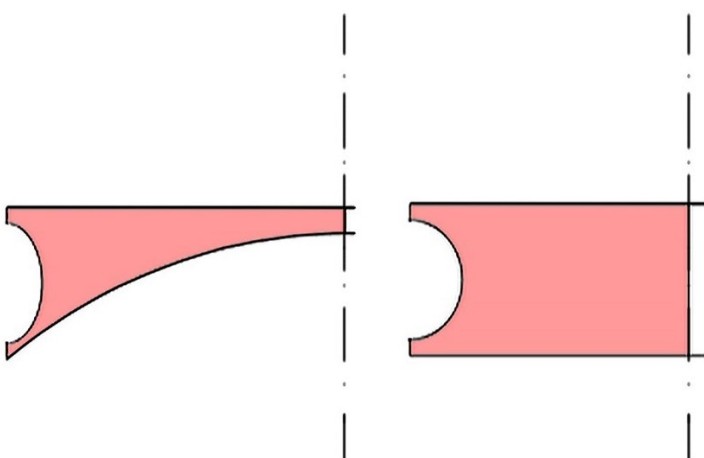

**Figure 8.** Trimmed void using PlaneTrimCurve.

In order to create equidistant voids on the remaining development of the beam, it is necessary to divide the entire length of the beam into *n-parts* of *n-equal* distance (Figure 9); the start and end-points of the length division thus represent the centers C of the beam voids, designed on the circumference arc passing through the points $P_{1'}$, $P_{2'}$, $P_{3'}$.

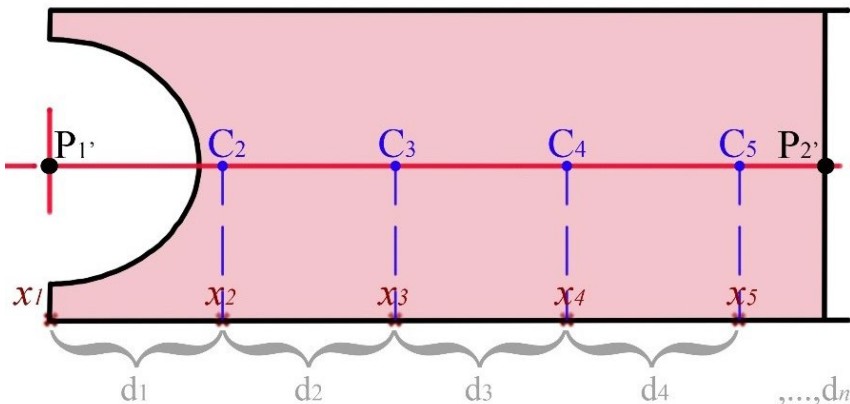

**Figure 9.** Division of the beam length.

The configuration of voids depends on the parameter $h_{m'}$ (Equation (3)); as the value of $h_{m'}$ decreases, the size of the first half-void increases. To prevent overlaps between the perimeters of the voids, it is necessary to shift the center of the second void:

$$x_{2'} = \frac{L}{n_d} + \frac{h_{m', \max} - h_m}{2} \tag{20}$$

where $n_d$ represents an arbitrary coefficient depending on the chosen number of voids.

Similarly to the parameter $a_1$, also the domain of $a_2$ must be defined in order to ensure an adequate distance $s$ between the voids (Figure 10):

$$(a_2 - s) \in [0.5a_1; a_1] \tag{21}$$

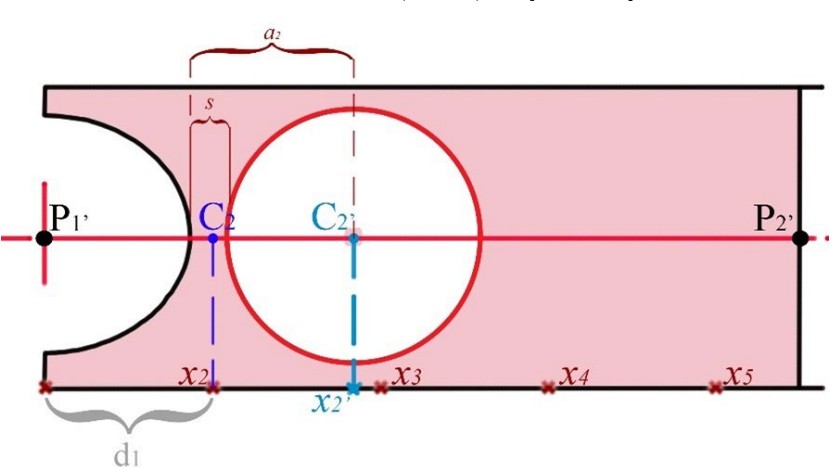

**Figure 10.** Void N.2.

The last parameter to be defined is the amplitude of $b_2$ belonging to the second void, as follows:

$$b_2 \in [0.1\, a_2\, ;\, a_2] \tag{22}$$

The third void of the structure is retrieved as the second one completing the possibilities of circular voids if the chosen final solution is a starting box with three circular voids (Figure 11).

In the examined case, the use of inclined uprights could provide a better distribution of stresses. Therefore, if the optimal solution contemplates ellipses, it is assumed that they can rotate around their center, counterclockwise, by an angle in the range of 0°–45° (if $a_n \neq b_n$) (Figure 12).

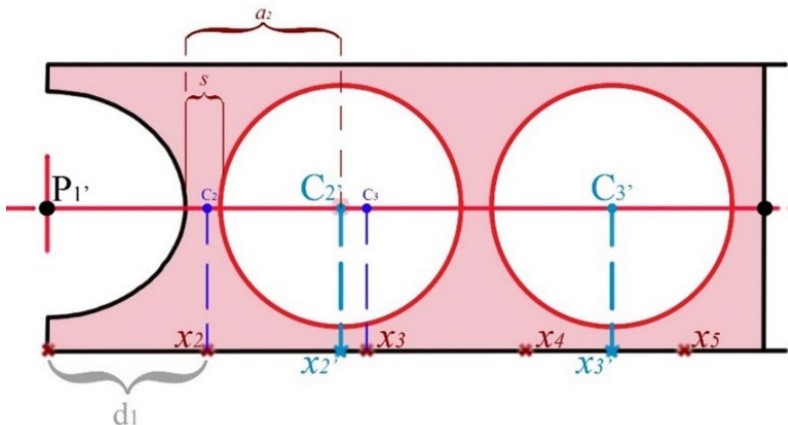

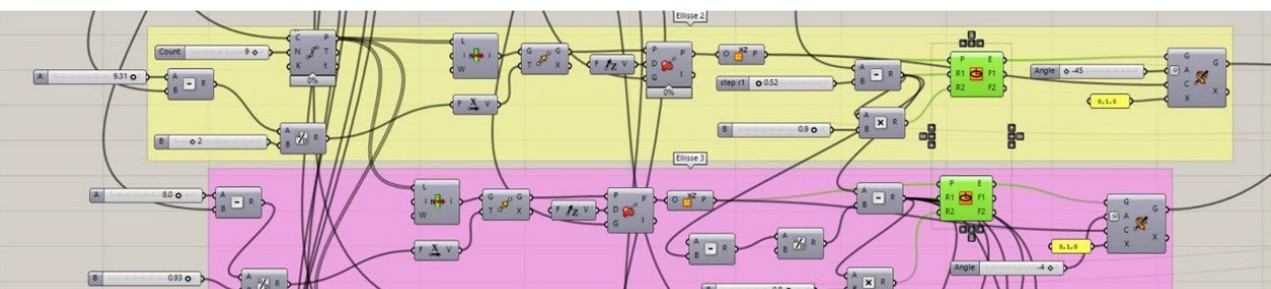

**Figure 11.** Void N.3.

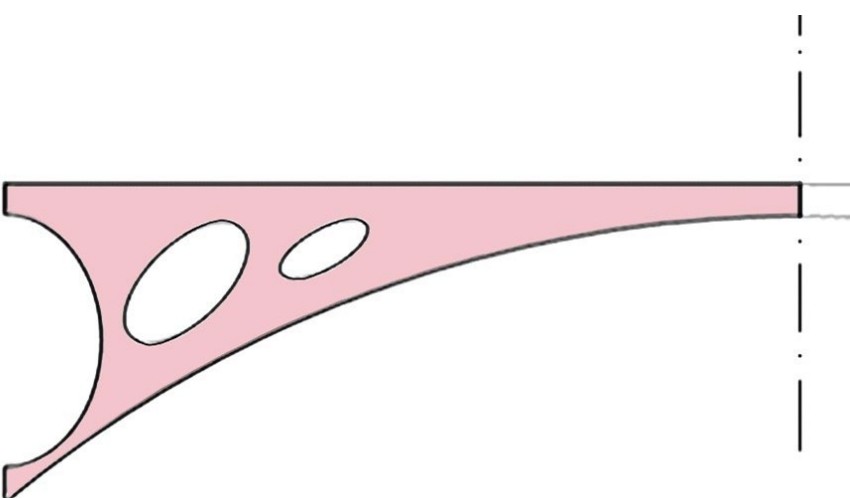

**Figure 12.** Rotated voids.

As shown in Figure 12, there is still material between the third void and the mid-span of the beam, that could be removed to obtain a lighter structure; by imposing the function as a Boolean test:

$$if(h_{m'} > 2/5h_e, y, 0) \tag{23}$$

additional voids can be introduced if the parameter $h_{m'}$ is larger than $2/5\ h_e$; in the case under examination, the number of voids can reach a maximum value equal to 3 + 3 (additional voids depending on Equation (23)) (Figure 13).

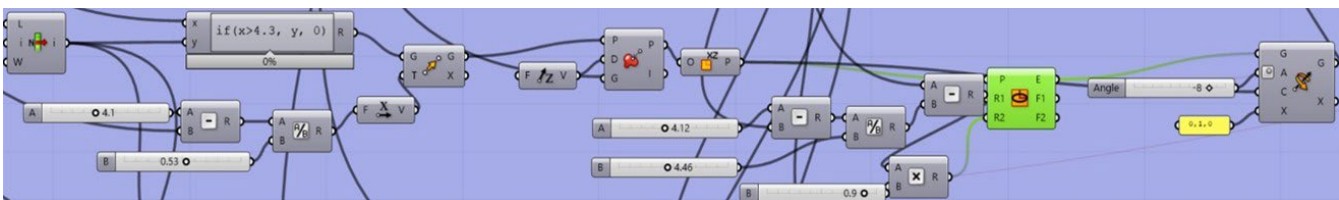

**Figure 13.** Boolean test in visual script to create additional voids if $h_{m'}$ is larger than 4 m (Equation (14)).

The horizontal radius $a_i$ and the corresponding vertical radius $b_i$ of successive voids are iteratively obtained by adopting a similar procedure. Joining the geometries in a single component by the *JoinCurves* tool, the geometric pattern of the examined problem is achieved (Figure 14), governed almost entirely by the parameter $h_{m'}$ (Equation (3)).

a
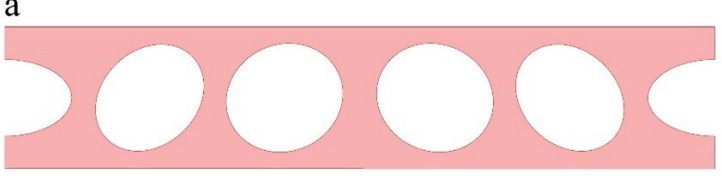

b
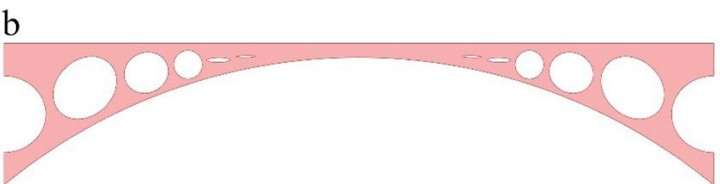

**Figure 14.** Joined geometry: (**a**) $h_{m'} = 0$; (**b**) $h_{m'} > 0$.

The transition from the geometrical model to the structural one is achieved by the *MeshBrep* tool (Karamba3d plug-in for Grasshopper) that allows switching from a geometric mesh to a finite element model (FEM), through a triangulation of parametric bidimensional elements.

### 2.2. Finite Element Analysis

In order to proceed with the FEA, starting from the retrieved model, it is necessary to specify the material and the cross-section dimensions. The *MeshtoShell* component allows for the retrieval of a shell model from given meshes as input and, at the same time, to define the cross-section of shell elements through the *CrossSection* component and the material by the *MatSelect* component. The cross-section is given by a *ShellConstant*, which allows setting the shell height and material with a constant cross-section. The selected material is concrete belonging to C45/55 class.

Since a more considerable thickness of the cross-section leads to a greater strength of the structural element, the problem concerning the cross-section is more controversial. Although the shape represents the main feature in the optimization problem (from both aesthetic and structural points of view), tensile stresses are herein minimized, so excluding the minimization of bending moments that would imply fewer thicknesses and costs. Moreover, in order to guarantee a concrete area able to accommodate the minimum reinforcement, a minimum thickness equal to 35 mm is introduced for the cross-section.

Constraints and loads represent the last two FEM input data.

As to constraints, at the two extremities of the beam, the points below are assumed clamped—considering them connected to the foundations—while a hinged roller constraint

is imposed on the points above, thus accounting for all possible longitudinal translations due to rheological and thermal effects (Figure 15).

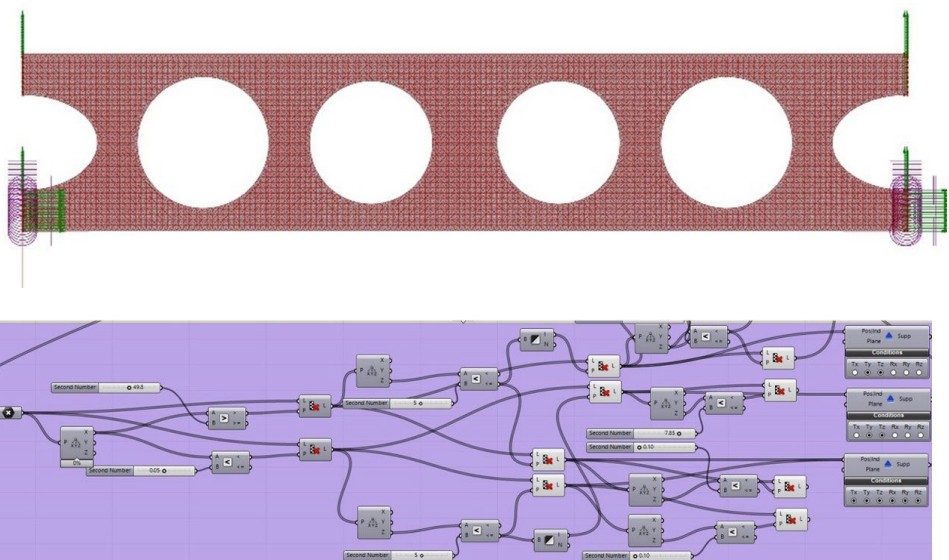

**Figure 15.** Supports implementation in visual scripting.

Due to the nature of the problem, in this preliminary study, the structure is subject just to its self-weight by adopting the *Load* command (Gravity).

Thus, after performing a linear static analysis of the FEM, the *AssembleModel* and *Analyse* components allow us to extract the values related to the mass and the displacements of the model. Furthermore, two different solutions are evaluated: the solid beam and the voided case with $h_{m'} \in [0, 0]$.

Similarly, the shell principal stresses, described as *Sig1-Val* and *Sig2-Val*, can be obtained from the *ShellVecResults* components. The results are summarized in Table 1 and Figure 16.

**Table 1.** FEA Results.

| Test Case | Displacement (cm) | Mass (Kg) | Sig1-Val [Tensile Stress] (kN/cm²) | Sig2-Val [Compressive Stress] (kN/cm²) |
|---|---|---|---|---|
| Solid beam | 0.33 | 437,500 | 0 to 1.119294 | −1.033997 to 0 |
| $h_{m'} \in [0, 0]$ | 1.13 | 191,044 | 0 to 1.429453 | −1.922351 to 0 |

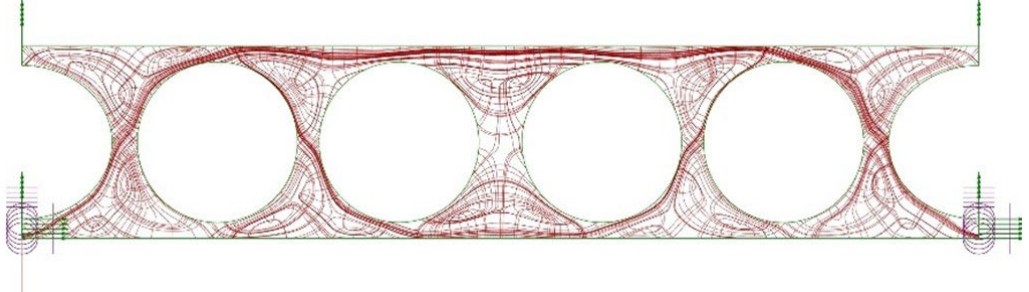

**Figure 16.** Sig1-Val and Sig2-Val path.

According to the Mohr circle model, the highest tensile stress *Sig1-Val* results equal to about $1.43 \text{ kN/cm}^2$, while compressive stresses reach a maximum value equal to $-1.92$ $\text{kN/cm}^2$ (Figure 16).

By disassembling the structural model, it is possible to retrieve the complete list of the mechanical characteristics of all the shell elements. So, after associating different colors to each element depending on the tension sign, a visual graph distribution of tensile and compressive stresses can be obtained in Rhinoceros 3D ambient (Figure 17). Figure 17 also provides an enlargement of the FEM mesh, showing its resolution, set to 0.150 m.

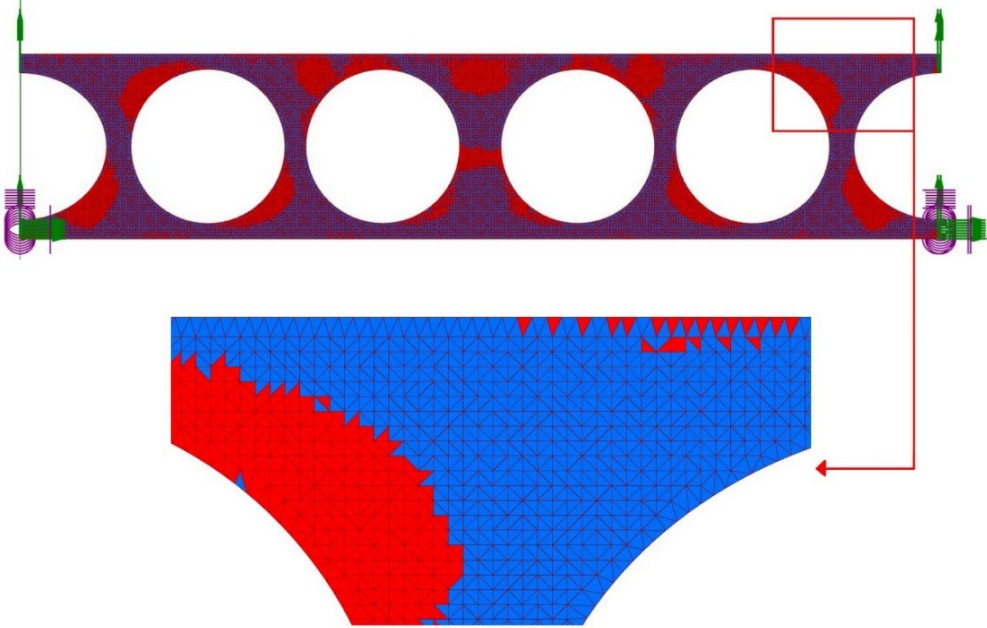

**Figure 17.** Tension (blue) (Sig1-Val) and compression (red) (Sig2-Val) path in the voided beam.

## 3. The Multi-Objective Optimization Problem

The optimal shape that minimizes both mass and tensile stresses of the voided beam is herein searched for. So, a MOOP is solved in order to find the best solution.

Moreover, it is imposed that tensile stresses do not exceed the concrete cracking limit, given by (Eurocode 2): $f_{ctm} = 0.3 f_{ck}^{2/3}$.

Therefore, the optimization problem can be formulated as follows:

*Find the*:

$$\min_{\{p_i\} \in P_{ad}} J_1, J_{2\max}(P_1, \ldots, P_9) \tag{24}$$

*By imposing the constraint*

$$J_{2max} \leq f_{ctm} \tag{25}$$

where $P_{ad}$ is a set of admissible parameters and $J_1$, $J_{2\max}$ represent the mass and the maximum tensile stress to be minimized, respectively [25].

Depending on the parameters included in Equation (15), the shape variation allows a possible solution set included in the Pareto optimal front. In the optimization process, the solver is able to find the best shape and dimensions of the voids to allow a more homogeneous distribution of stresses within the structure.

*Optimization Results—Octopus Solver (HypE Reduction)*

The Pareto optimal front in MOOPs represents a set of solutions that are non-dominated while being the best of all possible solutions. It means that a single solution cannot be simultaneously superior to all the other ones concerning all objectives. As a single goal improves in MOOP, another will get worse. Therefore, each solution of the Pareto set includes at least one objective that is inferior to another solution in that Pareto set, although both are superior to others in the rest of the search space [26]. This condition implies the necessity to define which is the best among all the feasible conditions.

In the case under examination, the Octopus solver adopts the search algorithm SPEA-2 to reach the convergence to the best global solution. SPEA2 uses a regular population and an archive. Starting with an initial population and an empty archive, the following steps are performed per iteration. First, all non-dominated population members are copied to the archive; any dominated individuals or duplicates are removed from the archive during this update operation. If the size of the updated archive exceeds a predefined limit, further archive members are deleted by a clustering technique that preserves the characteristics of the non-dominated front. Afterward, suitable fitness values are assigned to both archive and population members. The next step represents the mating selection phase where individuals from the union of population and archive are selected by means of binary tournaments. Finally, after recombination and mutation, the old population is replaced by the resulting offspring population [27].

After 250 generations, the solver selected a set of solutions, as shown the Figure 18. Three possible configurations are considered, in which the first solution (Test 1) shows the smallest values of tensile stresses among all feasible solutions, the third solution (Test 3) is the solution with the most negligible mass, and the second solution (Test 2) is characterized by mid-values with respect to the two edge solutions. The results after optimization (Figure 19), compared with the FEA of the solid beam, are summarized in Table 2. Considering the solid beam as a comparative test case, Table 3 summarizes the percentage variations of the chosen solutions.

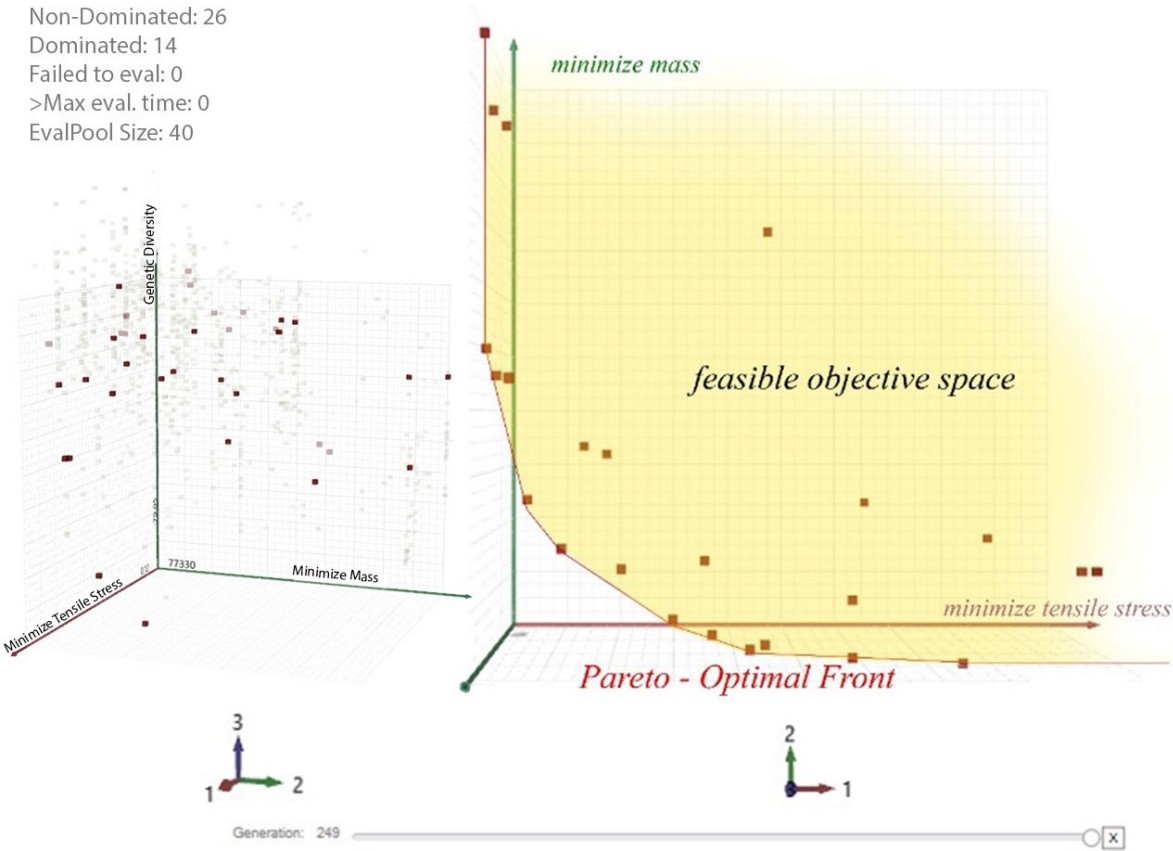

**Figure 18.** Pareto optimal front; in red, all feasible solutions.

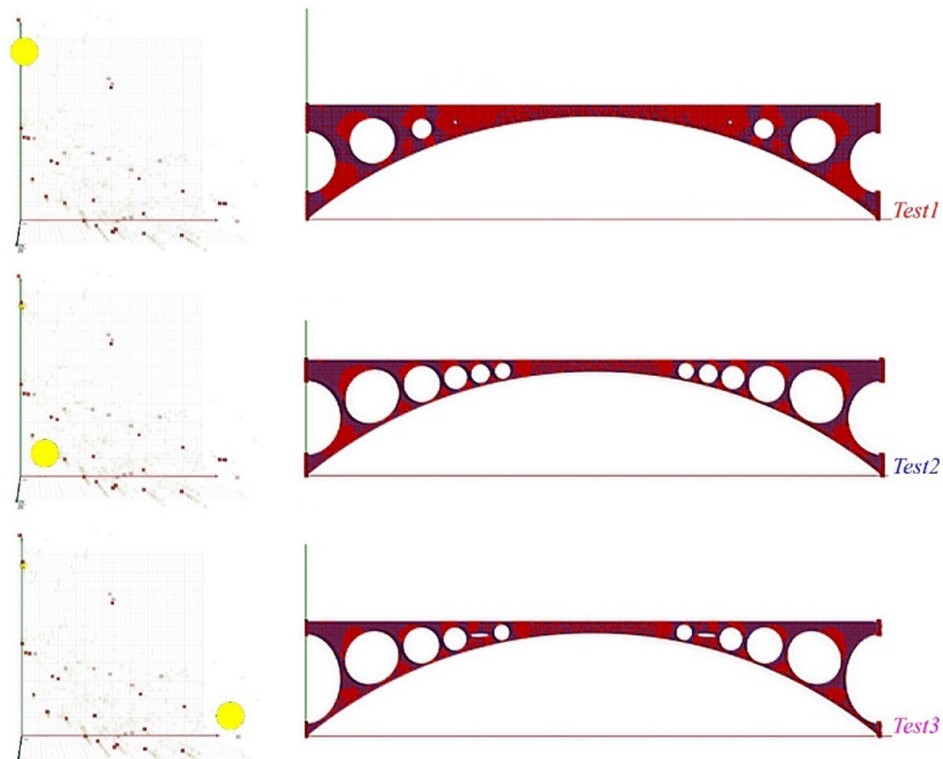

**Figure 19.** Three different optimization results retrieved in Pareto front.

**Table 2.** Optimization results.

| Test Case | Displacement (cm) | Mass (Kg) | Sig1-Val$_{max}$ [Tensile Stress$_{max}$] (kN/cm$^2$) |
|---|---|---|---|
| Solid beam | 0.33 | 437,500 | 1.119294 |
| Test 1 | 0.36 | 123,932.7 | 0.23137 |
| Test 2 | 0.51 | 86,963.8 | 0.283788 |
| Test 3 | 0.56 | 84,206.1 | 0.336129 |

**Table 3.** Percentage variation comparison.

| Test Case | Displacement (%) | Mass (%) | Sig1-Val$_{max}$ [Tensile Stress$_{max}$] (%) |
|---|---|---|---|
| Test 1 | +9.1 | −71 | −97.9 |
| Test 2 | +54 | −80.1 | −74.6 |
| Test 3 | +69.7 | −80.8 | −70 |

By observing, it emerges that the enhancement of one objective function corresponds to a worsening of the other. The following graph (Figure 20) shows the trend of the objective functions: it is worth noting that as the mass increases, the maximum tensile stress decreases.

Considering that concrete is characterized by good behavior in compression but a bad one in traction, for the purposes of this study, the most performing solution is given by Test Case 1. In fact, lower tensile stresses mean lower reinforcement, thus obtaining a double advantage: saving on concrete material and on steel reinforcement, which are economic and environmental benefits.

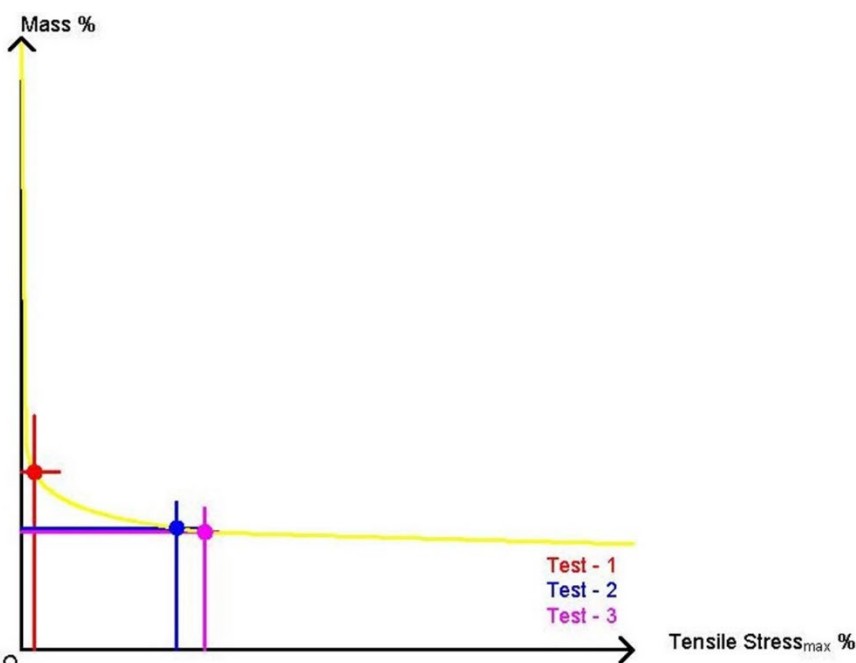

**Figure 20.** O.F. trend.

## 4. Form-Finding Using Kangaroo Engine

In this section, among the three solutions selected from Octopus results, the one closer to the main goals of the optimization is subjected to form-finding. Thus, Test Case 1, characterized by the lowest values of tensile stresses, is chosen for form-finding. The workflow of the last optimization process is summarized in Figure 21.

The Kangaroo solver adopts a new approach to form-finding, called the dynamic equilibrium method. It arrives at a static solution by using dynamic equilibrium equations. The structural element is decomposed in a particle-spring system, made up of particles and springs [28]. The particles are dimensionless points in space where all mass is concentrated and correspond to the mesh components. The springs connect particles to one another and are modeled as straight linear elastic bars. Typically, the mass of the particles represents the self-weight of the structural form. Once the form-finding process is initiated, the initial shape of the particle spring network is not in equilibrium and the forces begin to impact the particles. These forces are generated by the displacements of the springs from their rest length and the forces applied to the particles. Thus, the particles move through space until the forces acting on them are in equilibrium. At this point, the system essentially converges to a stable configuration.

So, the Kangaroo solver is a "Physical Laboratory" that iteratively moves the points to obtain the lowest sum of energies acting on all the points of the system. The length goal acts as a spring, following Hooke's law.

$$F = Kx \qquad (26)$$

which states that the force ($F$) needed to extend or compress a spring by a certain distance ($x$), linearly scales with respect to that distance, $k$ being a constant factor characteristic of the spring (i.e., its stiffness), and $x$ being small compared to the total possible deformation of the spring. The energy is zero at its rest length (length factor = 1) but increases if the spring is stretched or compressed.

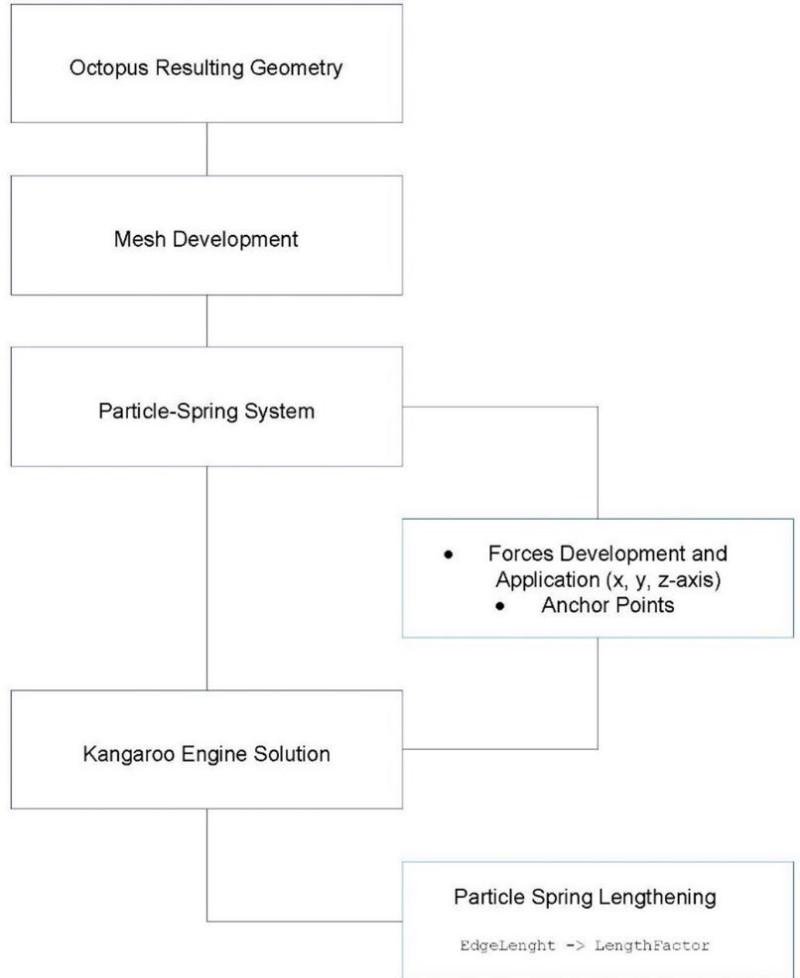

**Figure 21.** Kangaroo Engine workflow.

Different goals can be achieved by applying the form-finding method, such as defining energies on the basis of the geometric relations between the set of points they act on. Some properties (length, angle, pressure) are based on physical elastic behavior so that their strength can be precisely related to standard material properties and units. Other goals are purely geometric or physically based, such as the Hinge goal for shell bending, able to model physical behavior (only) qualitatively. This method can work as a geometric constraint solver when the goals do not conflict by making all the energies zero. However, in some cases, such as finding the deflection of a hanging cable, the length and load goals cannot be simultaneously satisfied—since one resists the other—so the solver finds the configuration with the least total potential energy. Nevertheless, this strategy is able to lead to numerically accurate elastic deformation models if the strengths are correctly set.

By using Kangaroo, tasks such as constraint solving, structural deformation modeling, and dynamic animation are all addressed by an energy minimization approach. The Kangaroo algorithm used to perform energy minimization can be seen as a form of dynamic relaxation (DR)—developed by Alistair Scott Day in 1974 [29]. The DR method consists in achieving equilibrium by combining all the forces acting on each point and repeatedly moving all the points in small steps until the force's balance is reached, and the movement stops. In the typical engineering applications of dynamic relaxation, the desired output is represented by the static equilibrium configuration, so that damping and mass values are changed to obtain stability and convergence rather than actual physical values. Kangaroo (Version2), based on DR, combines projections onto the zero-energy state of each goal [30].

*Kangaroo Solver Results*

In this section, the strategy of form-finding by Kangaroo Engine is applied to the case study herein analyzed. The elongation of the springs of the particle-spring system, as above described, is directly managed by fixing the line length into the *EdgeLengths* component, so setting the minimum length when compressed. After choosing the range 0.7/1.0 (real size) to regulate the elongation of the springs, it is necessary to impose suitable constraints to reproduce boundary conditions. By setting the *AnchorPoint* component, it is possible to fix the points in the x-z plane that overlap with the supporting points in the FEM. Finally, the solver is launched, reaching the convergence and so obtaining the solution described in Figure 22.

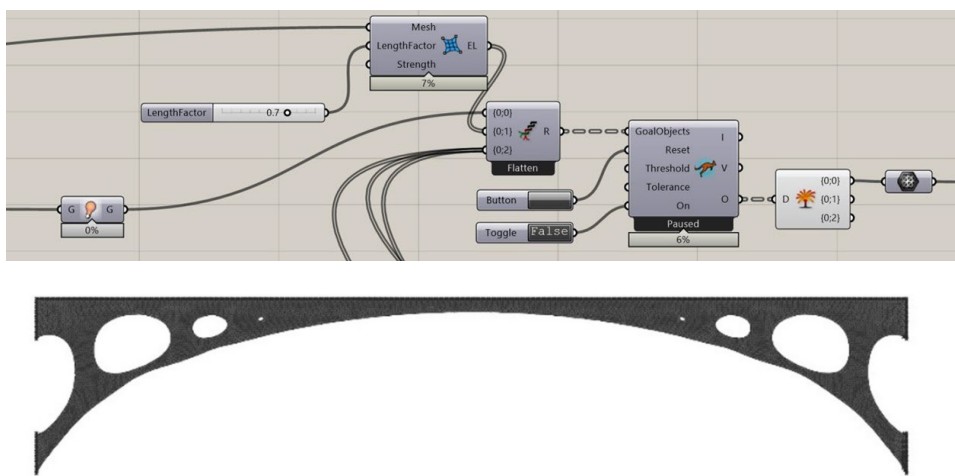

**Figure 22.** Kangaroo form-finding development and solution.

Connecting Karamba 3D (FEA) with Kangaroo Solver, the results summarized in Table 4 are retrieved.

**Table 4.** FEA results Kangaroo solver solution.

| Test Case | Displacement (cm) | Mass (kg) | Sig1-Val$_{max}$ [Tensile Stress$_{max}$] (kN/cm$^2$) |
|---|---|---|---|
| Kangaroo results | 0.59 | 93,973.8 | 0.143538 |

Due to the form-finding process, the mass is drastically reduced—from 123,932.7 kg to 93,973.8 kg—with a percentage variation equal to about −24.2%. Together with the mass reduction, an overall improvement in structural performance is obtained. In fact, by the Kangaroo solver, also a significant decrease in the maximum tensile stress *Sig1-Valmax*, up to about −39%, is achieved, with respect to the first step of the shape optimization. Moreover, by increasing the discretization of the entire mesh, the condition of total compression of the arc is reached (Figure 23).

The obtained solution can be considered optimal since, in a theoretical way, reinforcement is not necessary, and the risk of cracking is almost rare. So, if some percentage of reinforcement is anyway included according to code prescriptions, this solution allows reducing both its amount and concrete class, with consequent advantages in terms of costs and environmental impact. These benefits add up to the aesthetical ones, achieved thanks to the combination of shape-optimization and form-finding methods.

It is important to remark that, together with design and material costs, construction ones also have to be accounted for and affect the overall final cost of the structure; nevertheless, the adoption of constant curvature shapes, as the ones herein obtained by the final optimization through Kangaroo Solver, can be convenient from an economic point of view.

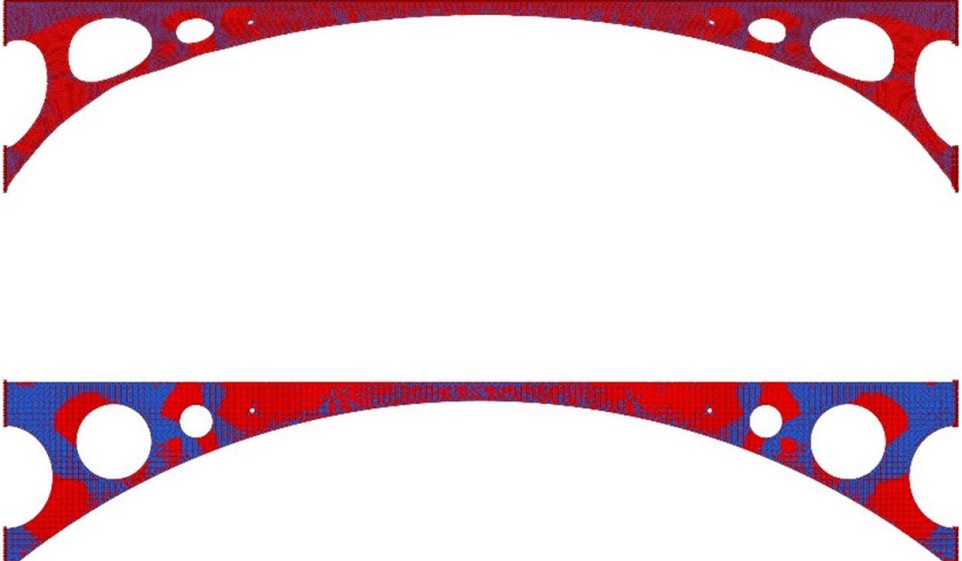

**Figure 23.** Comparison between shape-optimization result (bottom) and Kangaroo Solver.

### 5. Conclusions

In this paper, a new strategy for the preliminary optimization of a shell element was presented and tested using innovative tools that allow combining computational design with generative algorithms.

The main aim of the study was to understand the structural behavior of a new shape of the beam, an emptied voided beam, and to investigate the potentialities of this particular configuration, by minimizing both volume and tensile stresses.

By adopting the PSDO method, it is possible to reduce the time and complexity of calculation compared to more traditional methods, which often require the development of derived functions. Furthermore, by VP, accurate analyses can be implemented for the optimization of pre-processing and post-processing structural elements without necessarily resorting to complicated programming techniques (such as C++, C#, Phyton, etc.). The method also allows the integration, in the analysis, of functions and codes external to the Grasshopper environment, without elaborating too many complex models.

The proposed methodology can be summarized by the following steps: (*i*) preliminary analytical study to set the geometrical shape of the beam and of the voids; (*ii*) implementation of the geometrical model in Grasshopper; (*iii*) elaboration of a FEM by the Karamba3d plug-in; (*iv*) resolution of a MOOP by Octopus solver (HypE Reduction) and achievement of a Pareto optimal front of solutions; (*v*) selection of the solution closer to the main goals of the conceptual design and implementation of form-finding by Kangaroo Engine in order to further optimize the beam.

More precisely, by a double-step optimization (steps *iv* and *v*), it was possible to reduce the amount of material up to 80% compared to the initial solution of the solid beam, creating the possibility of pre-designing in a sustainable way with lower costs. Simultaneously, also a significant reduction in tensile stresses was obtained in the shell element ($-87.2\%$), leading to a decrease in the number of reinforcements and again of costs. These goals were reached under continuous control of the structural shape, thanks to the geometric constraints imposed in the preliminary geometry calculation phase.

The above results prove the efficacy of the proposed approach, based on direct communication between the geometry, modeled by a FEA, and the optimization problem. Finally, the presented method can be particularly useful in a preliminary design stage, to comprehend if the initial creative intuition of a construction shape also corresponds to good structural behavior, and to decide if it can undergo the successive, more detailed, design steps.

**Author Contributions:** Conceptualization, L.S., A.F. and R.G.; methodology, L.S.; software, L.S.; validation, L.S. and A.F.; formal analysis, L.S.; investigation, L.S. and A.F.; resources L.S.; data curation, A.F.; writing—original draft preparation, L.S.; writing—review and editing, A.F. and R.G.; visualization, L.S.; supervision, A.F., R.G. and A.M.; project administration, A.F. All authors have read and agreed to the published version of the manuscript.

**Funding:** This research received no external funding.

**Informed Consent Statement:** Not applicable.

**Conflicts of Interest:** The authors declare no conflict of interest.

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
