# Peer review of "Performative Structural Design Optimization: Generative Algorithm for a Preliminary Study of a Voided Beam"

_applsci, doi:10.3390/app12178663_

Round 1
Reviewer 1 Report
In this paper, a new method is presented for structural optimization purposes named “performative structural design optimization”. It is based on the “algorithm aided design” by searching for the most suitable shape using the multi-objective evolutionary algorithm. The optimization process is fulfilled by the hyper-volume estimation algorithm. The case study is selected to be a concrete beam that is voided to reduce its weight and volume without too much loss of stiffness and strength.
The paper needs a major revision based on the following:
1. Fig. 1 and its description should be moved to Sec. 2.
2. There is no literature review in this paper. Review of the related tasks for structural optimization is a must to put this study in the frame work.
3. Eqs. 2 and 3: Why is the specific shape of Eq. 2 selected? Why not presenting directly Eq. 3?
4. Eq. 4: Why choosing an arc shape?
5. Eq. 9: Why starting from an ellipse?
6. Fig. 12: It seems that structural design rules fo stiffness and strength has no place in this optimization algorithm!
7. Fig. 16: Please give a clear visual representation of the FEM mesh and its dimensons.0
8. Fig. 18, the Pareto-Optimal front: How is convergency to the best global solution guaranteed?
9. Sec. 4.1: Kangaroo Solver Results: “elongation of the springs”: What springs are meant?!
10. Table 4 and Fig. 23: While this the least-weight design, it cannot be the cheapest one and it is sure to increase the construction cost significantly compared with the conventional design costs. Please elaborate.
Reviewer 2 Report
The paper is well-written and may be of great interest to readers.
English checking and image processing should be carefully conducted.
Reviewer 3 Report
The article is devoted to the current topic of optimal design of building structures. A new software toolkit for automated design has been developed. The original is the combination in one process of meeting the requirements for strength, aesthetics and ergonomics of the designed structure.
Reviewer 4 Report
The manuscript has presented a new strategy for the optimization design of reinforced concrete shell structures. To this aim, many commercial softwares have been employed such as Rhinoceros 3D, Grasshopper, Karamba3D, Octopus, and Kangaroo. This study mostly can provide a useful technical manual for civil engineers in structural designs. Accordingly, I have some comments:
1. In Eq. (15), why does the maximum tension stress need to be included in the objective function, while it is set as an optimal constraint as well?
2. For topology optimization and form-finding of shells, relevant works should be addressed in the Introduction:
- Structural and Multidisciplinary Optimization 65 (4), 1-28
- Computers & Structures 259, 106695
3. In section 4, the form-finding process is only performed for Test 1. How about the performances of Test 2 and Test 3 using the form-finding stage?
A comparison and discussion need to be included.
Round 2
Reviewer 1 Report
The revised manuscript is acceptable.
Reviewer 4 Report
Thank you for your revising the manuscript.
The manuscript now can be published in its present form.